# Epidemiology and disease burden of sickle cell disease in France: A descriptive study based on a French nationwide claim database

Henri Leleu[1‡], Jean Benoit Arlet[2‡]*, Anoosha Habibi[3], Maryse Etienne-Julan[4], Mehdi Khellaf[5], Yolande Adjibi[6], France Pirenne[7,8], Marine Pitel[9], Anna Granghaud[9], Cynthia Sinniah[9], Mariane De Montalembert[10], Frédéric Galacteros[3]

**1** Public Health Expertise, Paris, France, **2** Internal Medicine Department, Sickle Cell Disease National Referral Center, Georges Pompidou European Hospital, Université de Paris, Paris, France, **3** Sickle Cell Referral Center, Internal Medicine Unit, IMRB Team 2, UPEC, Labex GRex, Henri Mondor Hospital, Créteil, France, **4** Referral Center for Sickle Cell Disease, Pointe à Pitre Hospital, Antilles University, Guadeloupe, France, **5** Paris Est University, INSERM U955, APHP, Emergency Department, Henri Mondor Hospital, Créteil, France, **6** FMDT Sos Globi, Henri Mondor Hospital, Créteil, France, **7** Etablissement Français du sang (EFS), Ile-de-France, Université Paris Est Créteil, Faculté de Médecine Créteil, Créteil, France, **8** Laboratory of Excellence GRex, INSERM U955, Créteil, France, **9** Pfizer, Paris, France, **10** Department of General Pediatrics and Pediatric Infectious Diseases, Reference Center for Sickle Cell Disease, Necker Hospital for Sick Children, Paris, France

‡ HL and JBA are co-first authors, they contributed equally to this work.
* henri.leleu@ph-expertise.com

**Data Availability Statement:** Data cannot be shared publicly as it contains potentially identifying and sensitive patient information, and access has been restricted by the National Commission for

## Abstract

### Context

Sickle cell disease (SCD) is a severe hematological disorder. The most common acute complication of SCD is vaso-occlusive crisis (VOC), but SCD is a systemic disease potentially involving all organs. SCD prevalence estimates rely mostly on extrapolations from incidence-based newborn screening programs, although recent improvements in survival may have led to an increase in prevalence, and immigration could account for a substantial number of prevalent patients in Europe. The primary objective of this study was to estimate SCD prevalence in France.

### Methods

A cross-sectional observational study was conducted using a representative sample of national health insurance data. SCD patients followed up in France between 2006 and 2011 were captured through hydroxyurea reimbursement and with the International Classification of Diseases (ICD-10) SCD specific code D570.1.2, excluding code D573 (which corresponds to sickle cell trait (SCT)). Nevertheless, we assumed that ICD-10 diagnosis coding for inpatient stays could be imperfect, with the possibility of SCT being miscoded as SCD. Therefore, prevalence was analyzed in two groups of patients [with at least **one (G1) or two (G2) inpatient stay**] based on the number of SCD-related inpatient stays in the six-year study period, assuming that SCT patients are rarely rehospitalized compared to SCD. The prevalence of SCD in the sample, which was considered to be representative of the French

Data Protection and Liberties (CNIL). Data access can by obtain through the Health Data Hub by following the procedures details on their website (https://www.health-data-hub.fr/).

**Funding:** This work received funding from Pfizer France in the form of a grant. Pfizer France had no role in the design of this study and did not have any role during its execution, analyses, interpretation of the data, or decision to submit results. Some authors have received honoraria from Novartis, Pfizer, Addmedica and BlueBirdBio detailed in the declaration of interest section of the manuscript. MP, AG, CS are employees of Pfizer France. The specific roles of these authors are articulated in the 'author contributions' section. MP, AG, CS reviewed and edited the manuscript but had no role in the design of this study and did not have any role during its execution, analyses, interpretation of the data, or final decision to submit the manuscript.

**Competing interests:** JBA reports honoraria and consultancy/expert testimony for Novartis and Pfizer. MM reports honoraria and consultancy/expert testimony for Novartis and Addmedica, and Board participation for BlueBirdBio; MEJ reports honoraria and consultancy/expert testimony for Novartis and Pfizer; AH reports honoraria and consultancy/expert testimony for Novartis and Addmedica, and Board participation for BlueBirdBio. This does not alter our adherence to PLOS ONE policies on sharing data and materials.

population, was then extrapolated to the general population. The rate of vaso-occlusive crisis (VOC) events was estimated based on hospitalizations, emergencies, opioid reimbursements, transfusions, and sick leave.

## Results

Based on the number of patients identified for G1 and G2, the 2016 French prevalence was estimated to be between 48.6 per 100,000 (G1) or 32,400 patients and 29.7 per 100,000 (G2) or 19,800 patients. An average of 1.51 VOC events per year were identified, with an increase frequency of 15 to 24 years of age. The average annual number of hospitalizations was between 0.70 (G1) and 1.11 (G2) per patient. Intensive care was observed in 7.6% of VOC-related hospitalizations. Fewer than 34% of SCD patients in our sample received hydroxyurea at any point in their follow-up. The annual average cost of SCD care is €5,528.70 (G1) to €6,643.80 (G2), with most costs arising from hospitalization and lab testing.

## Conclusion

Our study estimates SCD prevalence in France at between 19,800 and 32,400 patients in 2016, higher than previously published. This study highlights the significant disease burden associated with vaso-occlusive events.

## Introduction

Sickle cell disease (SCD) is a severe hematological disorder. It is caused by an inherited mutation of the beta globin (HBB) gene. Patients with the homozygous hemoglobin S (HbSS) mutation have sickle cell anemia, the most common and severe form of SCD. Other compound heterozygous genotypes are described with more variable clinical expressions (the most frequent being SC or S/β-thalassemia genotypes). People with a single HBβ gene (HbAS) have sickle cell traits rather than SCD, and are considered asymptomatic carriers [1, 2].

There is a large variation in symptom expression for SCD, both between individuals and, for a given individual, during a lifetime. The most common acute complication of SCD is vaso-occlusive crisis (VOC), which is usually accompanied by severe bone pain. SCD is also a chronic and systemic disease potentially involving all organs [3]. Acute complications of SCD include infection, acute chest syndrome, priapism, stroke, splenic sequestration, hepatobiliary complications, and acute renal failure [2, 4]. Chronic complications include avascular bone necrosis, pulmonary hypertension, heart failure, renal complications requiring dialysis, retinopathy, and leg ulcers [2, 4]. SCD patients are also at high risk of complications during pregnancy [5] and surgery. The heavy burden of SCD complications is responsible for premature death, with, in France, a median age at death of 36 years [6].

SCD treatment relies on both preventive and curative measures [2, 4, 7] to reduce the rate of acute and chronic complications alike. Treatment includes pain management for VOC (which frequently requires opioids), prevention, particularly pneumococcal vaccination, screening, and management of chronic complications. Disease-modifying therapies include treatment with blood transfusion and hydroxyurea. For patients with severe forms and benefiting from an identical HLA donor, usually a sibling, allogeneic hematopoietic stem cell

transplant is the only cure, with gene therapy under investigation [8]. Many ongoing trials are testing novel agents that could reduce both acute and chronic complications [8, 9].

SCD was acknowledged as a national health priority under the French Public Health Act of 2004. However, SCD prevalence and management is not well described in France. Most epidemiological assessment relies on the newborn screening program implemented in France in 1995 and systematized throughout the country in 2000. Based on this program, it is estimated that SCD affects 1 in 714 births [10], with most occurring in overseas departments and the Paris area. Newborn screening (targeted for SCD in mainland France) involves approximately 490 children per year. However, it is not possible to determine the prevalence of SCD patients in France by extrapolating newborn screening data. Indeed, a significant proportion of patients are of immigrant origin [11–13]. The recent increase in survival [14–16] due to earlier and better management also makes it difficult to extrapolate the prevalence from birth incidence as increased survival leads to more prevalence.

Therefore, the primary objective of this study was to estimate SCD prevalence in France. Secondary objectives include describing the number of hospitalizations and intensive care visits and calculating the average annual cost of care.

## Method

A cross-sectional observational study including French SCD patients followed up between 2011 and 2016 was conducted.

### Data source

SCD patients were identified in the permanent beneficiaries' sample (Échantillon Généraliste des Bénéficiaires: EGB) of the French health insurance information system [17]. It provides a 1/97th representative cross-sectional sample of the French population covered by National Health Insurance (NHI). The selection period was 2006–2016 but the follow-up period was 2011–2016. EGB covered about 95% of the population included in the NIH by 2016. Further description of the EGB data is available as S1 File.

### Patient identification

SCD patients were captured through hydroxyurea reimbursement, hospital care, and chronic condition (*Affection Longue Durée* [Long-Term Illness] ALD) status between 2006 and 2016. Based on expert opinions, we assumed that ICD-10 diagnosis coding for inpatient stays could be imperfect, with the possibility of sickle cell traits (code D573) being miscoded as sickle cell disease (D570: SCD with crisis, D571: SCD without crisis, D572: SCD heterozygous forms). Therefore, two groups of patients were defined to reduce uncertainty. **Group 1 (G1)** was defined as patients with at least **one inpatient stay** having an associated ICD diagnosis code of D57x (excluding D573: sickle cell traits) between 2006 and 2016, or ALD status with an associated ICD diagnosis code of D57 until 2016 or at least one hydroxyurea reimbursement between 2006 and 2016 (as described below). **Group 2 (G2)** was more strictly defined as patients with at least **two inpatient stays** having an associated ICD diagnosis code of D57x (excluding D573 for sickle cell traits) between 2006 and 2016, or ALD status with an associated ICD diagnosis code of D57 until 2016 or at least one hydroxyurea reimbursement between 2006 and 2016. It was assumed that this second group would include fewer misclassified patients as it required at least two inpatient stays associated with sickle cell disease, assuming that SCT patients are rarely rehospitalized compared to SCD [13, 18].

Hydroxyurea reimbursement included both Siklos® and Hydrea®. Because Hydrea® also has indications in other hematological myeloproliferative disorders, patients with Hydrea®

reimbursements that had no other SCD inclusion criteria and, were over 30 or had either ALD status associated with a hematological disorder were excluded.

As EGB coverage was extended from 85% of the French population in 2006 to 95% in 2016, some patients could move in or out of the database (e.g., becoming a student or self-employed). Therefore, patients with incomplete coverage between 2011 and 2016 were excluded so as not to underestimate VOC rates, complication rates, and resources used. This was considered more conservative than including partial follow-up as moving back to the main NHI group could be related to worsening of the disease-causing unemployment or extended sick leave.

Additionally, deceased patients remain in the EGB database. Therefore, patients deceased before January 1, 2016, were excluded as the primary objective was to estimate the 2016 prevalence of SCD in the French population. Inversely, patients born between 2006 and 2015 were included.

### Outcomes

Outcomes were estimated for each patient based on reimbursement data and inpatient stays.

VOC, treated as an inpatient or outpatient, was estimated based on hospitalization, emergency visits, opioid reimbursement, blood transfusion or red blood cells exchange apheresis or sick leave (for any reason). Hospitalization included all hospitalization related to a VOC-associated diagnosis, including SCD with crisis (D570), acute respiratory distress syndrome, pneumonia, pulmonary embolism, chest syndrome, lung infection, pulmonary thrombosis, priapism, neurological syndromes or pain (the ICD codes used are detailed in the S1 File). Blood transfusions or red blood cell exchange apheresis (routine transfusions, i.e., repeated regularly more than five times (or every month), were excluded as they were assumed to be related to long-term and not acute VOC treatment).

Because a patient's care during VOC could combine any of the above, events that were less than a month apart were considered to be related to a single crisis. A month was used as only months and years were available in the data.

All inpatient stays for included SCD patients were obtained. Chronic and acute complication rates were estimated based on the diagnosis associated with each inpatient stay.

Healthcare resources used and associated costs were also obtained, including inpatient stays, outpatient visits (GPs, pediatricians, orthopedists, nephrologists, dermatologists, internists, hematologists, nurses), laboratory tests (outpatient only), medical procedures (inpatient and outpatient), and treatments (folic acid, antalgics, anti-inflammatories, and vaccines). For healthcare resources used, a comparison group was created by matching SCD patients in the database to non-SCD patients (1:1) on age, sex, and place of residence.

### Resource use

Resource use was based on reimbursement data available for 2011 to 2016 for all patients and included all hospitalizations (with corresponding ICD-10 diagnosis), every reimbursement for prescription drugs purchased in community pharmacies (with corresponding ATC code), every procedure and lab tests performed as outpatients (with detailed procedure or lab test codes), every emergency visit and every consultation (with physician's specialty). For each reimbursement, the month and year were available.

### Analysis

Prevalence was estimated based on the prevalence of SCD in the EGB sample, which was considered to be representative of the French population, and then extrapolated to the general population.

Number and percentage were used to describe ordinal variables (binary or categorical). Average, median, standard deviation, quartiles, minimum, and maximum were used for quantitative variables. VOC rates and other complication rates were estimated per patient year of follow-up to reflect the difference in a follow-up for patients born between 2011 and 2016.

Similarly, resources used, and associated costs were estimated as averages per patient year of follow-up. For complication rates and resource use, we matched SCD-patients to non-SCD patients. Matched non-SCD patients were randomly selected from EGB. Patients were matched on age, sex and place of residence. Similar follow-up was used for matched non-SCD patients. McNemar's Chi$^2$ and paired t-tests were used to compare SCD patients to non-SCD patients. An alpha risk of 5% was used.

All analyses were performed with SAS 9.4 (SAS Institute, North Carolina, USA).

### Ethical consideration and informed consent

No informed consent was needed as all analyses were performed on anonymized data. The protocol was reviewed and approved by the Institut des Données de Santé (INDS), which manages data access for NHI. Approval by INDS waives the need for ethical clearance. This publication has been submitted to an associative opinion.

## Results

Fig 1 shows the inclusion flowchart. A total of 379 patients were identified from 703,261 beneficiaries based on group 1 criteria. Twenty-nine patients died between 2011 and 2016 and were excluded, leaving 350 patients. Of these patients, 93 had incomplete follow-up, leaving 257 patients included in group 1 (G1). Of these, 157 were included in group 2 (G2).

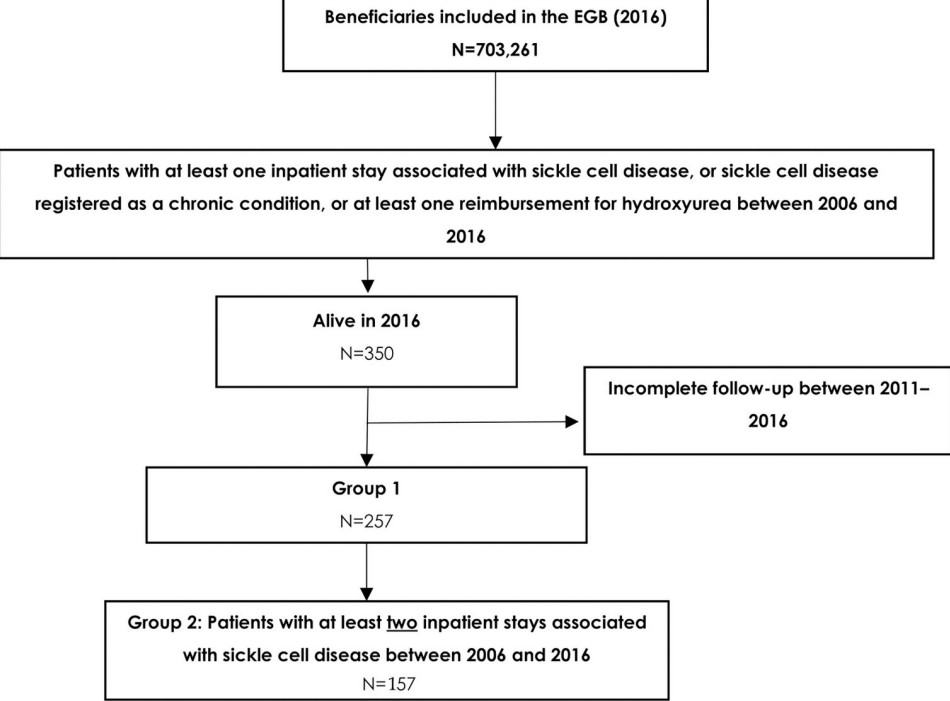

**Fig 1. Flow diagrams for the inclusion of SCD patients.** The selection period is 2006–2016 and the follow-up period 2011–2016.

**Table 1. Characteristics of SCD patients identified in the EGB database.**

| Characteristics | Group 1 N = 257 | Group 2 N = 157 |
|---|---|---|
| **Woman–n (%)** | 166 (64.6) | 93 (58.5) |
| **Median age (1–3 quartiles)** | 33 (17–48) | 31 (12–42) |
| **Registered as having chronic condition insurance (ALD)–n (%)** | 149 (58.0) | 118 (74.2) |
| **Residence** | | |
| **Metropolitan France–n (%)** | 221 (86.0) | 136 (86,6) |
| **Paris area–n (%)** | 119 (46.3) | 81 (51.6) |
| **Overseas departments–n (%)** | 36 (14.0) | 21 (13.3) |
| **Vaso-occlusive crises (number per patient per year)** | 1.51 | 1.90 |
| **SCD-related hospitalizations (number per patient per year)** | 0.40 | 0.63 |
| **Opioid analgesics (outpatient) (number per patient per year)** | 0.55 | 0.66 |
| **Emergency visits not followed by hospitalization (number per patient per year)** | 0.36 | 0.38 |
| **Sick leave (number per patient per year)** | 0.12 | 0.11 |
| **Erythrocyte exchanges (number per patient per year)** | 0.05 | 0.08 |
| **Transfusions (number per patient per year)** | 0.02 | 0.04 |
| **Incidence of hospitalization (number per patient per year)** | 0.70 | 1.11 |
| **Including VOC[1] (/ patient / year)** | 0.40 | 0.63 |
| **Including transfusions/erythrocyte exchanges (/ patient / year)** | 0.07 | 0.12 |
| **Including renal failure or dialysis (/ patient / year)** | 0.12 | 0.20 |
| **Average (SD) number of transfusions or erythrocyte exchanges per year** | 0.14 (0.70) | 0.21 (0.88) |
| **Bone marrow transplants–n (%) [2007–2016]** | 4 (1.6) | 4 (2.5) |
| **Hydroxyurea–n (%)** | 60 (23.3) | 60 (37.6) |

SD: Standard deviation.

[1] VOC was estimated based on: hospitalization related to a VOC-associated diagnosis, including sickle cell anemia with crisis (D570), acute respiratory distress syndrome, pneumonia, pulmonary embolism, chest syndrome, lung infection, pulmonary thrombosis, priapism, neurological syndromes or pain (the ICD codes used are detailed in the S1 File); emergency visits; opioid reimbursements; blood transfusions or red blood cell exchange apheresis (routine transfusions, i.e. repeated regularly more than five times (or every month), were excluded as they were assumed to be related to long-term and not acute VOC treatment); sick leave (for any reason)

When including all patients with at least one inpatient stay related to SCD between 2006 and 2016 (G1), the 2016 French prevalence was estimated to be 48.6 per 100,000 or 32,400 patients. Using the stricter definition for G2, with at least two inpatient stays related to SCD, French prevalence was estimated to be 29.7 per 100,000 or 19,800 patients.

The characteristics of G1 and G2 patients are presented in Table 1. Patients had a median age of 33 in G1 and 31 in G2 with 64.6% and 58.5% females respectively. Between half and two thirds were registered as having ALD status. Most patients (G1: 86.0%, G2: 86.8%) reside in metropolitan France, including half (G1: 46.3%, G2: 50.9%) in the Paris area, and 14.0% (G1) to 13.2% (G2) in overseas departments.

Based on the study methodology, patients had on average 1.51 (G1) and 1.90 (G2) VOC per year, including those treated as inpatient or outpatient (with inpatient stays, outpatient opioid reimbursements, and emergency visits representing the majority of identified VOC situations); sick leave and isolated transfusions accounted for a minority. Between 23.3% (G1) and 37.6% (G2) of patients had at least one hydroxyurea reimbursement between 2011 and 2016. The average annual number of transfusions, including those associated with VOC, was 0.07 to 0.12

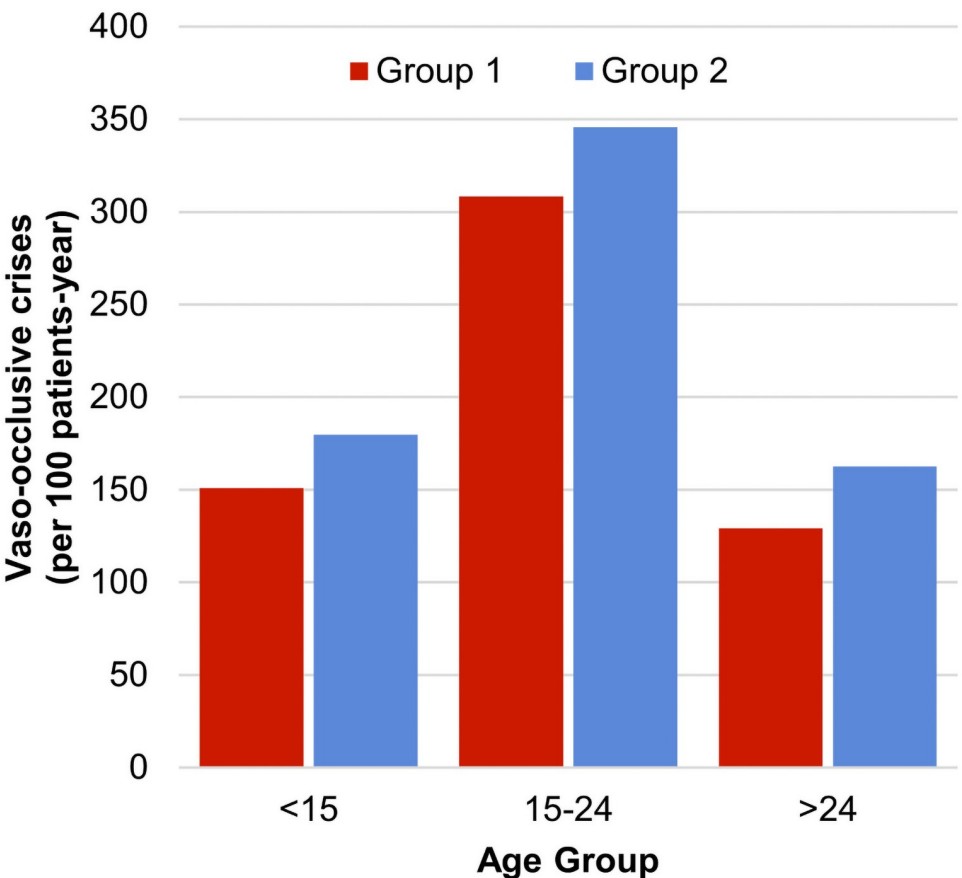

**Fig 2. Frequency of vaso-occlusive crises by age group.**

per patient. In the sample, only four patients received autologous bone marrow transplants during the period.

The average annual number of hospitalizations was between 0.70 (G1) and 1.11 (G2) per patient, with VOC representing 0.40 to 0.62 hospitalizations per patient per year. By comparison, in matched non-SCD patients with the same age in the national database, the average annual hospitalization rate was 0.01 per patient per year. Other main diagnoses associated with inpatient stays were renal failure (including dialysis) and transfusion. Excluding VOC, renal failure, and transfusion, SCD patients had 0.10 to 0.17 hospitalizations per patient per year for SCD-related complications. Details are provided in S1 File.

Of the 604 VOC-associated hospitalizations identified in the database for included SCD patients, 248 (41.1%) were day hospitalizations, 76 (12.6%) lasted a single night, and 280 (46.3%) at least 2 nights. Average length of stay (excluding day hospitalization) was 4.3 (standard deviation: 4.0) days (median of 3 days; quartiles 2–5). Of hospitalization lasting one night or more, 320 (89.9%) followed an emergency visit. Of all hospitalizations, 27 (7.6%) required intensive care for a median duration of 2 days (quartiles 2–8).

Fig 2 presents the distribution of VOC frequency by age. The frequency increases between 15 and 24 years (3.08/3.45 for G1/G2 between 15 and 24 versus 1.50/1.80 before 15), a difficult age for disease management where patients go through adolescence and leave pediatric care for adult care.

**Table 2. Average annual healthcare resource used per patient.**

| Healthcare Resource (n/year) | SCD Patients (Group 1) N = 257 | Non-SCD Patients N = 257 | p |
|---|---|---|---|
| **VISITS (annual number of visits per patient)** | | | |
| Outpatient Visits with GPs | 5.57 | 4.13 | 0.0005 |
| Outpatient Visits with pediatricians | 1.54 | 0.54 | 0.0189 |
| Outpatient Visits with hematologists or internists | 0.56 | 0.05 | <0.0001 |
| Outpatient Visits with nephrologists | 0.18 | 0.00 | 0.0304 |
| Nurse Visits | 26.71 | 2.26 | 0.0422 |
| Emergency Visits | 0.51 | 0.21 | <0.0001 |
| **LAB TESTS* (annual number of lab tests per patient)** | | | |
| Hemogram | 2.73 | 0.95 | <0.0001 |
| Reticulocytes | 1.08 | 0.03 | <0.0001 |
| Irregular Blood Group Antibodies | 0.59 | 0.19 | 0.0372 |
| Liver function | 3.67 | 1.02 | <0.0001 |
| Kidney function (Blood electrolytes, Creatinine, Microalbuminuria, Proteinuria) | 3.62 | 1.07 | <0.0001 |
| Alkaline phosphatase, calcium, phosphorus | 1.49 | 0.31 | <0.0001 |
| Iron metabolism | 1.46 | 0.36 | <0.0001 |
| Fasting glucose | 1.21 | 0.64 | <0.0001 |
| **MEDICAL PROCEDURES (annual number of medical procedures per patient)** | | | |
| Electrocardiogram (ECG) | 0.54 | 0.09 | <0.0001 |
| Eye exam (Ophthalmoscopy, Tomography) | 1.02 | 0.16 | <0.0001 |
| Chest X-Ray | 0.37 | 0.14 | <0.0001 |
| Brain MRI | 0.11 | 0.03 | <0.0001 |
| Doppler ultrasound (Transcranial, Transthoracic) | 0.94 | 0.07 | <0.0001 |
| **TREATMENTS (box/year) (annual number of boxes of treatment per patient)** | | | |
| Folic Acid | 3.03 | 0.06 | <0.0001 |
| Non-opioid Analgesic (Paracetamol, Aspirin, Ibuprofen, Nefopam) | 6.39 | 2.93 | <0.0001 |
| Opioid (Codeine, Tramadol, Morphine) | 2.08 | 0.28 | <0.0001 |
| Hydroxyurea | 0.27 | 0.00 | 0.0014 |
| **VACCINES** | | | |
| Flu | 0.19 | 0.04 | <0.0001 |
| Meningococcus | 0.12 | 0.00 | <0.0001 |
| Pneumococcus | 0.08 | 0.00 | <0.0001 |

Comparison between SCD (group 1) and non-SCD patients matched on age, sex, and place of residence.

Table 2 presents the average annual healthcare resource per patient for group 1 SCD patients compared to non-SCD patients matched on age, sex, and place of residence. On average, SCD patients had significantly more outpatient visits with GPs or pediatricians than non-SCD patients, with 7.1 visits per year compared to 4.7. They had significantly more lab tests. This included not only more blood cell counts per year (2.7 versus 1.0) but also liver function, kidney function, pancreatic function, and iron or bone metabolism. Patients also had on average 0.6 irregular blood group antibody screenings per year. SCD patients had significantly more medical procedures, including chest X-ray, electrocardiogram (ECG) likely related to VOC or screening for the chronic effects of SCD, eye examination, Doppler ultrasound, and brain MRI. Finally, SCD patients had significantly more pharmacy dispensing for drugs related to VOC, including non-opioid analgesics (6.4 versus 2.9 dispenses per year) and opioids (2.1 versus 0.3), to the prevention of infection complications (vaccines), and to the long-term treatment of SCD with folic acid and hydroxyurea.

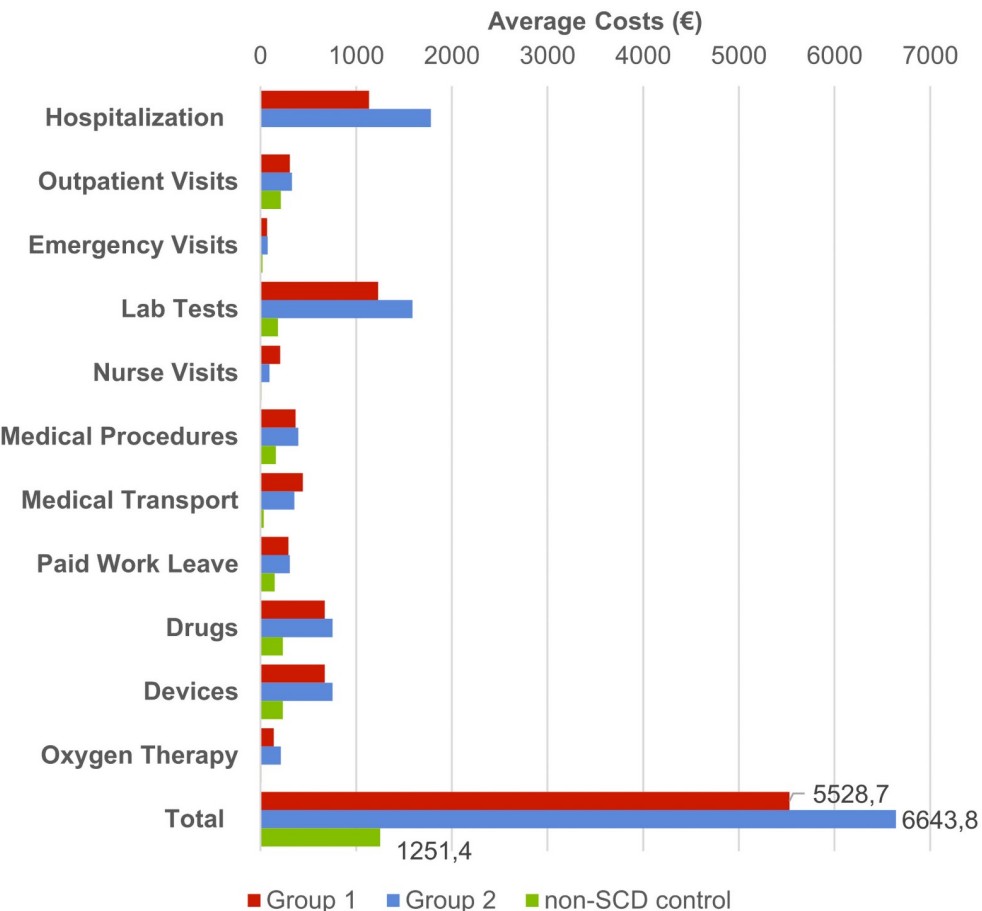

**Fig 3. Annual average cost of care by healthcare resources for group 1 and group 2 patients.** \*non-SCD control is based on data for the control group of group 2 patients.

Fig 3 presents the annual cost of care by healthcare resources for group 1 and group 2 SCD patients compared to non-SCD patients. Overall, the annual average cost of SCD care is between €5,528.70 (group 1) and €6,643.80 (group 2) compared to €1,251.40 for matched non-SCD patients. Most costs arise from hospitalization and lab testing, with drugs and medical devices the third and fourth contributors to total costs. Compared to non-SCD patients, this represents a €4,176.2 and €5,392.5 increase (p<0.001) in average annual care costs for SCD patients.

## Discussion

This study reports the prevalence, complication rate, resource use, and cost of care for SCD patients in France based on real-world data. Using the EGB data, we estimate that SCD prevalence in France is between 19,800 and 32,400 patients in 2016. Median age was 31 to 33, significantly less than the median age of 41 in the country's general population. This analysis confirms the significant burden of SCD, with patients presenting on average 1.51 to 1.90 VOC per year, including 0.40 to 0.63 hospitalized VOC. Compared to matched non-SCD patients, SCD patients had, as expected, significantly higher use of non-opioid (6.4 versus 2.9 packs per year) and opioid analgesics (2.1 versus 0.3). SCD patients also have more frequent hospitalizations (0.70 to 1.11 per patient per year compared to 0.01 in matched non-SCD patients), and a

significantly greater number of emergency visits, outpatient visits, blood tests or medical procedures than the general population, associated with both the acute and chronic effects of the disease.

The use of the EGB data presented major advantages. This is a longitudinal study based on real-world data with follow-up data since 2004 for more than 700,000 individuals representative of the French population. It is therefore possible to make prevalence estimates that can be extrapolated to the French population. However, this data also has some limitations that must be taken into account to interpret our results properly. First and foremost, the EGB data does not contain any independently confirmed diagnosis, therefore patient identification must rely on a combination of chronic disease (ALD) status, associated inpatient diagnosis, and specific treatments, lab tests or procedures, each of which has its biases and shortcomings.

Hydroxyurea can also be used to identify SCD patients as Siklos® is only indicated in SCD. However, some patients received or still receive Hydrea®, which also has an indication in malignant hematological diseases affecting older patients, requiring us to exclude patients aged over 30 with a Hydrea® reimbursement and no other SCD inclusion criteria. Furthermore, fewer than 34% of SCD patients in our sample received hydroxyurea at any point in their follow-up.

Finally, hospitalization provides the most sensitive method of identifying SCD patients as the long follow-up available made it unlikely, given the frequency of VOC and of day hospitalization for transfusion or chronic complications screening, that SCD patients would never be hospitalized over a ten-year period in France. However, this sensitivity means that there is a potential risk of false positives in our sample, i.e., patients misclassified as SCD. That is why we chose to include two definitions of SCD patients based on having either one or more or two or more SCD-associated hospitalizations. As with diagnostic tests, repeating the "test" reduces the risk of miscoding [12, 19, 20]. We excluded over 100 patients (39%) by using a more restrictive definition (a least two hospitalizations in the six-year period). Group 1 has a higher median age and is strongly biased toward females, which does not fit with an autosomal recessive disorder, even taking into account a potential survival bias toward females [14]. In-depth analysis of the hospitalizations associated with patients excluded from group 2 shows that they are strongly related to pregnancy. A possible assumption is that some female patients with sickle cell traits, already known or discovered during pregnancy follow-up, are miscoded as SCD. Another assumption is that pregnancy is a high-risk circumstance for the manifestations of SCD; these women are very often transfused while the hydroxyurea is stopped. Many SCD patients who are not very active (SC genotype, etc.) become so during pregnancy. This biases group 1 toward a less severe profile. Additionally, it is likely that some patients with SC and Sβ+ (SCD genotypes), which can present with less severe symptoms, and so lower rates of inpatient stays, have been excluded from group 2 by the >1 inpatient stay cut-off. The SC and Sβ+ genotypes could represent up to 20–30% of SCD patients in France [12, 13, 21, 22]. Therefore, while group 1 probably includes sickle cell traits, it is likely that group 2 excludes some SC and Sβ+ patients, leaving the SCD prevalence and results somewhere in the middle between groups 1 and 2.

An additional limitation of the EGB sample is the exclusion of university students and immigrants not covered by NHI. Therefore, our sample excludes a portion of patients aged between 15 and 25, as can be seen from the age distribution of patients (S1 File), leading to a potential underestimate of prevalence and of average VOC frequency as the child to adult transition is associated with the highest VOC rate [23, 24]. This might also explain why the median age observed is slightly higher than previous published studies [25, 26]. Similarly, given the high prevalence of SCD in immigration from sub-Saharan Africa [12, 13, 27], and with sub-Saharan Africa representing 43% of the beneficiaries of AME [28]. a specific health insurance

scheme for immigrants not covered by NHI, prevalence is further underestimated by this exclusion. Overall, it is likely that the prevalence rate of 29.7 per 100,000 or 19,800 patients for group 2 is a very conservative estimate. France seems to be the European country with the highest prevalence of SCD patients, followed by the UK with 14,000 patients (2018) [29].

Despite these limitations, our results are consistent with previously reported data. VOC rates are consistent with reported rates [30–34]. Hospitalization characteristics are also similar to previously reported data [35], as was the rate of hydroxyurea prescribed.

Lastly, concerning the annual average cost of care, we note that the costs related to lab tests are equivalent to hospitalization costs. Both expenses are significant cost drivers. In comparison, the average annual cost of care for a diabetic patient represents €2,169, half the cost of a sickle cell patient [between €5,528.70 (G1) and €6,643.80 (G2) in our study]. However, this annual cost of care remains much lower than for other conditions such as hemophilia (€11,046) or cystic fibrosis (€35,527) [36].

In conclusion, our study estimates that SCD prevalence in France in 2016 is between 19,800 and 32,400 patients, higher than previously published and one of the higher prevalence in Europe. It also confirms the heavy disease burden associated with SCD, with frequent and severe complications leading to a significant increase in healthcare use.

## Supporting information

**S1 File. EGB description.**
(DOCX)

## Acknowledgments

We thank the Assurance Maladie from providing access to the data.

## Author Contributions

**Conceptualization:** Henri Leleu, Jean Benoit Arlet, Maryse Etienne-Julan, Mehdi Khellaf, Yolande Adjibi, France Pirenne, Mariane De Montalembert, Frédéric Galacteros.

**Formal analysis:** Henri Leleu.

**Methodology:** Henri Leleu, Jean Benoit Arlet, Anoosha Habibi, Maryse Etienne-Julan, Mehdi Khellaf, Yolande Adjibi, France Pirenne, Mariane De Montalembert, Frédéric Galacteros.

**Supervision:** Jean Benoit Arlet.

**Validation:** Jean Benoit Arlet, Anoosha Habibi, Maryse Etienne-Julan, Mehdi Khellaf, Yolande Adjibi, France Pirenne, Mariane De Montalembert, Frédéric Galacteros.

**Writing – original draft:** Henri Leleu, Jean Benoit Arlet.

**Writing – review & editing:** Anoosha Habibi, Maryse Etienne-Julan, Mehdi Khellaf, Yolande Adjibi, France Pirenne, Marine Pitel, Anna Granghaud, Cynthia Sinniah, Mariane De Montalembert, Frédéric Galacteros.

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
