## [Decision Letter · Decision Letter 0]

3 Mar 2021

PONE-D-21-00775

Epidemiology and disease burden of sickle cell disease in France: a descriptive study based on a French nationwide claim database

PLOS ONE

Dear Dr. Leleu

Thank you for submitting your manuscript to PLOS ONE. After careful consideration, we feel that it has merit but does not fully meet PLOS ONE’s publication criteria as it currently stands. Therefore, we invite you to submit a revised version of the manuscript that addresses the points raised during the review process.

Please submit your revised manuscript by 2nd April 2021. If you will need more time than this to complete your revisions, please reply to this message or contact the journal office at plosone@plos.org. Please include the following items when submitting your revised manuscript:

We look forward to receiving your revised manuscript.

Kind regards,

Ambroise Wonkam, MD, PhD

Academic Editor

PLOS ONE

Journal Requirements:

2. In the ethics statement in the manuscript and in the online submission form, please provide additional information about the patient records/samples used in your retrospective study, including the date range (month and year) during which patients' medical records/samples were accessed.

"This work was funded by Pfizer France."

We note that one or more of the authors have an affiliation to the commercial funders of this research study : Pfizer.

3.1. Please provide an amended Funding Statement declaring this commercial affiliation, as well as a statement regarding the Role of Funders in your study. If the funding organization did not play a role in the study design, data collection and analysis, decision to publish, or preparation of the manuscript and only provided financial support in the form of authors' salaries and/or research materials, please review your statements relating to the author contributions, and ensure you have specifically and accurately indicated the role(s) that these authors had in your study. You can update author roles in the Author Contributions section of the online submission form.

3.2. Please also provide an updated Competing Interests Statement declaring this commercial affiliation along with any other relevant declarations relating to employment, consultancy, patents, products in development, or marketed products, etc.  

5. Please upload a copy of Supporting Material which you refer to in your text on pages 11, 14 and 19.

Reviewers' comments:

Reviewer's Responses to Questions

**Comments to the Author**

1. Is the manuscript technically sound, and do the data support the conclusions?

Reviewer #1: Yes

Reviewer #2: Yes

2. Has the statistical analysis been performed appropriately and rigorously? 

Reviewer #1: Yes

Reviewer #2: Yes

3. Have the authors made all data underlying the findings in their manuscript fully available?

Reviewer #1: No

Reviewer #2: Yes

4. Is the manuscript presented in an intelligible fashion and written in standard English?

Reviewer #1: Yes

Reviewer #2: Yes

5. Review Comments to the Author

Reviewer #1: Reviewer Comments to Author:

I reviewed the manuscript entitled "Epidemiology and disease burden of sickle cell disease in France: a descriptive study based on a French nationwide claim database"

The manuscript addresses an important epidemiological questions that may help in the care of patients with Sickle cell disease.

Generally the manuscript is well written and of scientific merit

Some minor revisions will improve the readability of the manuscript

Pease enlighten me on the choice of two groups (G1) and (G2). I would expect you to pick count the patient in your prevalence study regardless of the number of visits(hospitalization) they have had in the National Health Insurance claim database. You could just pick any SCD patient who have had at least one hospitalization event. This would make the estimated prevalence a single number rather than the current report which give a range for group 1 and group 2. You would still need to state the limitation of using this method of estimation.

I am interested to know how many records were excluded because of International Classification of Diseases (ICD-10) code D573 (which corresponds to sickle cell trait (SCT) in your dataset. Or other exclusion reasons.

Page 6: The statement the "aim of the study was to describe SCD prevalence in France"; the word describe can be interchanged with estimate

On page 7: The French longform for the abbreaviation (EGB) could be given along with the English description (permanent beneficiaries’ sample!)

A little description how the health insurance the 5% not covered by the NHI is covered, would help the reader to understand the excluded population. Would the majority of these 5% be immigrants from Sub-Saharan Africa where the prevalence of SCD is much higher than France?

The the first occurrence of the abbreviation ALD should include both the French and English equivalence on its first occurrence. Now ALD is defined in more than one place in the document. Do this check for other abbreviations as well.

Page 8: The paragraph starting with "Hydroxyurea reimbursement included both Siklos® and Hydrea®" need to be checked for clarity!

How was the data extraction process, were there patients that were excluded from the dataset because they database did not include (had blank) ICD-10? This would explain the data incompleteness and accuracy of the prevalence estimate from the database.

Page 8 and 9:try to write the how the VOC was estimated in a narrative format rather than outline, do the same for other section that were outlined in bullet format

Page 9: The statement "All inpatient stays between 2006 and 2016 for included SCD patients were obtained. Chronic and acute complication rates were estimated based on the diagnosis associated with each inpatient stay" is repeating as the duration of study was specified above this line.

Page 9: Which algorithm was built to estimate inpatient and outpatient stays? how does it work?

Page 9: The follow-up section is too brief or ectopic

Page 9: Describe how matched non-SCD patients were selected.

Page 9: Ethical consideration statement need clarity. What is the long form of IND? was a waiver for ethical clearance given by any Institutional Review Board?

Page 10: in the statement "All analyses were performed with SAS 9.4 (SAS Institute)". What is SAS(? (include state and country)

Page 11-13 (Results) please check clarity especially where brackets () are used.

.

Page 13: (Discussion) The second sentence can be shortened to report prevalence separate from age

Page 14: Hydoxyurea cab be written starting with lower case letter.

Compare your SCD, prevalence, VOC rate etc with other countries in Europe or global. Do French SCD patients experience less severe crises than other countries?

Reference number 25 can be written in proper letter case like other references. Please correct

Reviewer #2: Leleu H et al reported a cross-sectional study on prevalence of sickle cell disease (SCD) in France based on national insurance data from 2006 to 2011. The Authors intersected SCD population by ICD as well as reimbursement for hydroxyurea (HU). Leleu et al evaluated the rate of VOC indirectly by # of hospitalization, access to the ED, opioid reimbursement, transfusion. Using this approach, the Authors found a prevalence of SCD to higher than that expected. This highlights the increasing burden of the disease as well as the presence of sub-set of SCD patients possibly undertreated or with limited access to SCD comprehensive SCD centers.

Major points

• Method section is too long. Please move part of the description to supplementary methods.

• Please add some more comments on HU

• Please rewrite sentence on genotypes is too speculative

• Figure legends need to be implemented.

Minor

• Please check the manuscript for typos.

6. PLOS authors have the option to publish the peer review history of their article (what does this mean?). If published, this will include your full peer review and any attached files.

Reviewer #1: **Yes: **Raphael Zozimus Sangeda

Reviewer #2: No

---

## [Author Response · Author response to Decision Letter 0]

11 Jun 2021

March 9, 2021,

Ambroise Wonkam, MD, PhD, 

Academic Editor, PLOS ONE,

Thank you very much for the opportunity to submit a revised version of the manuscript PONE-D-21-00775 “Epidemiology and disease burden of sickle cell disease in France: a descriptive study based on a French nationwide claim database." We want to thank the reviewers for their comments, which have allowed us to strengthen the manuscript. A point-by-point response is included below, and the corresponding changes have been made in the manuscript and highlighted in yellow in a separate file.

The revised manuscript contains no data, patient information, or other material or results that have been published or are in press or submitted elsewhere. 

We look forward to hearing from you and thank you in advance for considering this contribution.

Best regards,

Henri Leleu, M.D., M.P.H., Ph.D.,

On behalf of the authors

 

Reviewer #1: 

I reviewed the manuscript entitled "Epidemiology and disease burden of sickle cell disease in France: a descriptive study based on a French nationwide claim database." The manuscript addresses an important epidemiological question that may help in the care of patients with Sickle cell disease. Generally, the manuscript is well written and of scientific merit.

Response: We thank the reviewer for his interest in our manuscript.

Pease enlighten me on the choice of two groups (G1) and (G2). I would expect you to pick count the patient in your prevalence study regardless of the number of visits(hospitalization) they have had in the National Health Insurance claim database. You could just pick any SCD patient who has had at least one hospitalization event. This would make the estimated prevalence a single number rather than the current report which gives a range for group 1 and group 2. You would still need to state the limitation of using this method of estimation.

Response: The two groups were chosen because the National Health Insurance claim database is a claim database and not an epidemiological cohort. Thus, diagnosis is not clinically confirmed, but is rather inferred based on reimbursement of sickle cell treatments, long-term allowances (ALD) for sickle cell and hospitalization with sickle cell as the main and secondary diagnosis. Unfortunately, for the latter, the amount hospitals are reimbursed is similar whether sickle cell disease or sickle cell traits are coded. This can lead to possibly miscoding a sickle cell trait as sickle disease in practice. This is apparent when you look at the characteristics of patients included in G1 and not in G2 (G1’) with 73% of women and a mean age of 39.1 years. Our assumption, confirmed by the clinicians, is that the overrepresentation of women in these patients is because sickle cell trait is likely diagnosed during pregnancy and is probably miscoded as sickle cell disease. As the probability of repeated miscoding is low, we decided to use at least two hospitalizations as a strong certainty of sickle cell disease (G2) while patients with a single hospitalization with no other criteria (G1’) were possibly miscoded sickle cell traits. We chose to present data for G1 and G2 as we assumed that the correct epidemiological value stands somewhere in the middle. This is discussed in the discussion section of the manuscript: “

“We excluded over 100 patients (39%) by using a more restrictive definition (a least two hospitalizations in the six-year period). Group 1 has a higher median age and is strongly biased toward females, which does not fit with an autosomal recessive disorder, even taking into account a potential survival bias toward females.14 In-depth analysis of the hospitalizations associated with patients excluded from group 2 shows that they are strongly related to pregnancy. A possible assumption is that some female patients with sickle cell traits, already known or discovered during pregnancy follow-up, are miscoded as SCD. Another assumption is that pregnancy is a high-risk circumstance for the manifestations of SCD; these women are very often transfused while the Hydroxyurea is stopped. Many SCD patients who are not very active (SC genotype, etc.) become so during pregnancy. This biases group 1 toward a less severe profile. Additionally, it is likely that some patients with SC and Sβ+ (SCD genotypes), which can present with less severe symptoms, and so lower rates of inpatient stays, have been excluded from group 2 by the >1 inpatient stay cut-off. The SC and Sβ+ genotypes could represent up to 20–30% of SCD patients in France.12,13,21,22 Therefore, while group 1 probably includes sickle cell traits, it is likely that group 2 excludes some SC and Sβ+ patients, leaving the SCD prevalence and results somewhere in the middle between groups 1 and 2.”

I am interested to know how many records were excluded because of International Classification of Diseases (ICD-10) code D573 (which corresponds to sickle cell trait (SCT) in your dataset. Or other exclusion reasons.

Response: Of the 340 patients that has at least one hospitalization during the study period with an associated sickle cell disease or sickle cell trait, 44 (13%) had only a sickle cell trait diagnosis and were excluded.

Page 6: The statement the "aim of the study was to describe SCD prevalence in France"; the word describe can be interchanged with estimate

Response: We made the corresponding change.

On page 7: The French longform for the abbreviation (EGB) could be given along with the English description (permanent beneficiaries’ sample!)

Response: We added the French longform for the abbreviation.

A little description how the health insurance the 5% not covered by the NHI is covered would help the reader to understand the excluded population. Would the majority of these 5% be immigrants from Sub-Saharan Africa where the prevalence of SCD is much higher than France?

Response: The sentence “NHI covered about 95% of the population by 2016” on page 7 was actually not correct and should have read, “EGB covered about 95% of the population included in the NIH by 2016”. This has been corrected in the manuscript. This is a technical issue that is related to the fact that the management of the NIH coverage for some groups of the population is subcontracted to different public or not for profit entities for historical reasons, and these entities are not fully integrated to the information system of the NHI, and thus are not fully integrated to the EGB that is extracted from the NIH information system. This is particularly relevant for students. Historically, students had their own health insurance systems that were eventually integrated to the NHI but are not yet fully integrated to the information system. This is really a technical issue but was deemed relevant for the manuscript as this slightly biases the sample because it excludes some students. However, we wanted to avoid going into in-depth details of the complexities of the NIH information system.

The first occurrence of the abbreviation ALD should include both the French and English equivalence on its first occurrence. Now ALD is defined in more than one place in the document. Do this check for other abbreviations as well.

Response: We had added the English equivalence. The literal translation would be “long-term illness”. 

Page 8: The paragraph starting with "Hydroxyurea reimbursement included both Siklos® and Hydrea®" need to be checked for clarity! How was the data extraction process, were there patients that were excluded from the dataset because they database did not include (had blank) ICD-10? This would explain the data incompleteness and accuracy of the prevalence estimate from the database.

Response: The paragraph was rewritten to be more explicit:

“patients with Hydrea® reimbursements that had no other SCD inclusion criteria and were over 30 or had either ALD status associated with a hematological disorder were excluded.”

All patients with Hydrea® reimbursements were initially selected. Patients with another SCD inclusion criteria, either ALD or hospitalization, were included. Some patients have an Hydrea® reimbursement without ALD or hospitalization. ICD-10 diagnosis is only available either because patients have ALD or have been hospitalized for SCD. ALD is not systematic in SCD patients, in fact, as shown in table 1 of the manuscripts, only between 74% (G2) and 58% (G1) of SCD patients have ALD. This is because patients can get similar benefits through other coverages (such as free complementary health insurance for low-income patients, or complementary health insurance through work), and ALD is associated with social stigma as the fact that a patient has the ALD status can be disclosed to the employer for example (although without the diagnosis). Similarly, as discuss previously, milder SCD could never be hospitalized with all VOD treated as outpatients. Thus, for some patients, there are legitimate reasons not to have ICD-10 diagnosis and it was not considered as a missing data. For these patients that had an Hydrea® reimbursement without ALD or hospitalization, to avoid including other indications of Hydrea® and thus bias the estimates, we excluded patients that had a hematological disorder diagnosis or were older than 30.

Page 8 and 9: try to write the how the VOC was estimated in a narrative format rather than outline, do the same for other sections that were outlined in bullet format.

Response: The corresponding section was rewritten in a narrative format.

Page 9: The statement "All inpatient stays between 2006 and 2016 for included SCD patients were obtained. Chronic and acute complication rates were estimated based on the diagnosis associated with each inpatient stay" is repeating as the duration of study was specified above this line.

Response: “between 2006 and 2016” was removed.

Page 9: Which algorithm was built to estimate inpatient and outpatient stays? How does it work?

Response: What is referred to as “this algorithm” is just how VOC were estimated, detailed in the paragraph above. For clarity we remove the sentence “This algorithm was built to estimate inpatient and outpatient stays” and added, “VOC, treated as inpatient or outpatient, was estimated…” in the previous section.

Page 9: The follow-up section is too brief or ectopic

Response: No follow-up was done per se, and the analysis was retrospective. We have replaced follow-up with “resource use” for more clarity. Resource use was based on the available reimbursement data. As the French National Health Insurance is a unique system, reimbursement data is comprehensive. We have added the following paragraph to better detailed the data available for resource use:

“ Resource use was based on reimbursement data available for 2011 to 2016 for all patients and included all hospitalizations (with corresponding ICD-10 diagnosis), every reimbursement for prescription drugs purchased in community pharmacies (with corresponding ATC code), every procedure and lab tests performed as outpatients (with detailed procedure or lab test codes), every emergency visit and every consultation (with physician’s specialty). For each reimbursement, the month and year were available.”

Page 9: Describe how matched non-SCD patients were selected.

Response: As described in the data source section, the EGB is a representative sample of the French population. Matched non-SCD patients were randomly selected in the EGB sample after matching on age, sex and place of residence. We added the following to the manuscript:

“For complication rates and resource use, we matched SCD-patients to non-SCD patients. Matched non-SCD patients were randomly selected from EGB. Patients were matched on age, sex and department of residence.”

Page 9: Ethical consideration statement needs clarity. What is the long form of IND? Was a waiver for ethical clearance given by any Institutional Review Board?

Response: INDS is the National Institute for Health Data, that give clearance to use the EGB data. This clearance waives any need for an IRB or Ethical Clearance. The EGB received a national ethical clearance. As long as a project complies with EGB guidelines (which is checked by INDS) then the project automatically benefits from the EGB clearance.

Page 10: in the statement "All analyses were performed with SAS 9.4 (SAS Institute)". What is SAS(? (include state and country)

Response: SAS is one of the leading statistical software in the word. The state and country have been added to the manuscript.

Page 11-13 (Results) please check clarity especially where brackets () are used.

Response: We have made some minor corrections (highlighted in yellow in the manuscript) to improve clarity.

Page 13: (Discussion) The second sentence can be shortened to report prevalence separate from age

Response: We have broken the sentence in two.

Page 14: Hydoxyurea cab be written starting with lower case letter.

Response: This has been corrected.

Compare your SCD, prevalence, VOC rate etc with other countries in Europe or global. Do French SCD patients experience less severe crises than other countries?

Response: No comparison is done because we have not identified at the time of the publication any reference for prevalence in other countries in Europe.

Reference number 25 can be written in proper letter case like other references. Please correct

Response: This has been corrected.

 

Reviewer #2: Leleu H et al reported a cross-sectional study on prevalence of sickle cell disease (SCD) in France based on national insurance data from 2006 to 2011. The Authors intersected SCD population by ICD as well as reimbursement for hydroxyurea (HU). Leleu et al evaluated the rate of VOC indirectly by # of hospitalization, access to the ED, opioid reimbursement, transfusion. Using this approach, the authors found a prevalence of SCD to higher than that expected. This highlights the increasing burden of the disease as well as the presence of sub-set of SCD patients possibly undertreated or with limited access to SCD comprehensive SCD centers.

Response: We thank the reviewer for his interest in our manuscript, and for the time spent reviewing.

Major points

• Method section is too long. Please move part of the description to supplementary methods.

Response: We agree with the reviewer that the method section is long. However, we believe that all the information is important to correctly understand the manuscript. The use of claim database for epidemiological study is uncommon and warrants a clear description of what has been done. However, to shorten the method section, the description of the EGB data has been moved to supplementary materials.

• Please add some more comments on HU

Response: We have rewritten the paragraph to be more explicit.

• Please rewrite sentence on genotypes is too speculative

Response: We have further explained this point in response to reviewer #1 comment on the choice of two groups (G1) and (G2). We have strong certainty that our interpretation is correct, however, it remains a speculation, as confirmation would require to either go back to the patients’ clinical record, which is impossible with EGB by design, or interview every French hospital to know about their coding practices which is not feasible.

• Figure legends need to be implemented.

Response: Figure legends have been added.

Minor

• Please check the manuscript for typos.

Response: The manuscript has been checked for typos.

---

## [Editor Report · Decision Letter 1]

17 Jun 2021

Epidemiology and disease burden of sickle cell disease in France: a descriptive study based on a French nationwide claim database

PONE-D-21-00775R1

Dear Dr. Henri Leleu,

We’re pleased to inform you that your manuscript has been judged scientifically suitable for publication and will be formally accepted for publication once it meets all outstanding technical requirements.

Kind regards,

Ambroise Wonkam, MD, PhD

Academic Editor

PLOS ONE
---

## [Editor Report · Acceptance letter]

30 Jun 2021

PONE-D-21-00775R1 

Epidemiology and disease burden of sickle cell disease in France: a descriptive study based on a French nationwide claim database 

Dear Dr. Leleu:

I'm pleased to inform you that your manuscript has been deemed suitable for publication in PLOS ONE. Congratulations! Your manuscript is now with our production department. 

Kind regards, 

on behalf of

Professor Ambroise Wonkam 

Academic Editor

PLOS ONE